# Metabolomics Response to Drought Stress in *Morus alba* L. Variety Yu-711

**DOI:** 10.3390/plants10081636

**Published:** 2021-08-09

**Authors:** Michael Ackah, Yisu Shi, Mengmeng Wu, Lei Wang, Peng Guo, Liangliang Guo, Xin Jin, Shaocong Li, Qiaonan Zhang, Changyu Qiu, Qiang Lin, Weiguo Zhao

**Affiliations:** 1School of Biology and Technology, Jiangsu University of Science and Technology, Sibaidu, Zhenjiang 212018, China; shiyisu1297@126.com (Y.S.); 199310015@stu.just.edu.cn (M.W.); Wangleiwendy@126.com (L.W.); gp52941@sina.com (P.G.); 192310021@stu.just.edu.cn (L.G.); jinxin9502@126.com (X.J.); ShaocongLi1@126.com (S.L.); Loer9725@126.com (Q.Z.); 2Sericulture Research Institute, Guangxi Zhuang Autonomous Region, Nanning 530007, China; Changyuqiu2008@163.com (C.Q.); gxlq67@163.com (Q.L.)

**Keywords:** untargeted approach, metabolites, drought stress, mulberry, LC-MS

## Abstract

Mulberry is an economically significant crop for the sericulture industry worldwide. Stresses such as drought exposure have a significant influence on plant survival. Because metabolome directly reflects plant physiological condition, performing a global metabolomic analysis is one technique to examine this influence. Using a liquid chromatography-mass spectrometry (LC-MS) technique based on an untargeted metabolomic approach, the effect of drought stress on mulberry Yu-711 metabolic balance was examined. For this objective, Yu-711 leaves were subjected to two weeks of drought stress treatment and control without drought stress. Numerous differentially accumulated metabolic components in response to drought stress treatment were revealed by multivariate and univariate statistical analysis. Drought stress treatment (EG) revealed a more differentiated metabolite response than the control (CK). We found that the levels of total lipids, galactolipids, and phospholipids (PC, PA, PE) were significantly altered, producing 48% of the total differentially expressed metabolites. Fatty acyls components were the most abundant lipids expressed and decreased considerably by 73.6%. On the other hand, the prenol lipids class of lipids increased in drought leaves. Other classes of metabolites, including polyphenols (flavonoids and cinnamic acid), organic acid (amino acids), carbohydrates, benzenoids, and organoheterocyclic, had a dynamic trend in response to the drought stress. However, their levels under drought stress decreased significantly compared to the control. These findings give an overview for the understanding of global plant metabolic changes in defense mechanisms by revealing the mulberry plant metabolic profile through differentially accumulated compounds.

## 1. Introduction

Crop losses have increased in recent decades due to changing weather patterns linked to climate change, and climate models forecast that droughts, floods, and severe temperatures will become more prevalent [1]. Combined with rising food demand, this condition is expected to threaten global food security. Drought remains one of the most damaging environmental stresses, as it inhibits plant growth and decreases productivity [2]. Drought stress has severe morphological, physiological, and biochemical consequences for plants, including reduced photosynthesis, disrupted cell elongation and division, and loss of cell turgor [3]. It also affects crop gene expression, distribution, yield, and quality by preventing plants from absorbing additional nutrients [4,5]. Therefore, a deeper understanding of tree crop responses to water deficit is critical to screen drought-tolerant varieties or genotypes that can respond to future climate scenarios in order to combat this effect.

Mulberry (*Morus alba* L., Moraceae) is cultivated for its leaves and fruits in most provinces of China [6]. In addition, the only food source for domestic silkworms (*Bombyx mori* L.) is mulberry leaves, making them a valuable tree species in the sericulture industry [7]. The Morus genus contains 68 species that grow in a wide range of climates from temperate to tropical, mainly in Asia, with one-third (24) of the species naturally occurring in China [8,9]. Thus, besides its traditional use in silkworm rearing, mulberry is a promising pioneer tree species on marginal lands [8,10].

Mulberry trees require an adequate water supply to reach rapid growth, which is one reason why large-scale mulberry plantations have historically been developed in humid areas of China. Conversely, new mulberry plantations have spread to areas where water is scarce to meet economic and ecological demands [8]. As a result, drought events occurred in these plantations on a seasonal or regional basis, resulting in lower yield and deterioration of mulberry tree quality [11]. Abiotic stress tolerance is a polygenic trait in plants that includes signal transduction pathways and interactions between multiple genes [12]. Nonetheless, drought or soil moisture deficit stress is a negative abiotic factor that significantly reduces mulberry foliage yield. It is estimated that about 20% of the world’s land surface is expected to be in drought at any time [13]. In order to counteract drought stress, drought adaptation traits such as effective water conservation, broader and deeper root systems, enhanced photosynthetic production, water use efficiency, and cellular tolerance must be introgressed into mulberry [9].

Drought, alkalinity, salinity, and frost-tolerant viable mulberry varieties have been developed using conventional breeding practices based on morphological and physiological phenotyping [14]. Wild mulberry species and germplasm resources, such as *M. serrata* and *M. laveigata*, have a robust growth pattern, higher resilience to adverse environmental circumstances, and silkworm palatability [15]. However, abiotic stress adaptive traits from wild species are yet to be incorporated into commercially viable mulberry variants [15].

Increasing food supply significantly in an unstable environment and passing this information to farmers in the required timeframe is essential [16]. Achieving this has been done gradually through breeding [17] and biotechnology [18], employing novel methods to develop more resilient plants to abiotic stress. In recent years, much progress has been achieved in plant metabolomics applied to abiotic stress research [19,20,21]. Many plants have used their extensive metabolic homeostasis to respond to various abiotic stressors by producing diverse primary and secondary metabolites [21]. Plant metabolites and associated pathways may be affected by abiotic stresses [3,22]. Vital metabolic pathways, including photosynthesis, sugar synthesis, tricarboxylic acid (TCA) cycle, glycolysis, amino acids, and hormone synthesis, could all be involved in plant responses to drought stress [3]. Metabolomics has primarily focused on the organic molecular compounds (metabolites) found in or formed by organisms, tissues, and cells [23,24]. Metabolomics is now widely used in plant science as an essential biotechnological method for various molecular biology studies. It encompasses a wide range of analytical techniques for identifying organic molecular metabolomic materials. These techniques include metabolic fingerprinting, metabolite profiling, targeted analysis, gas chromatography-mass spectrometry (GC-MS), liquid chromatography-mass spectrometry (LC-MS), and nuclear magnetic resonance (NMR) [25]. In addition, metabolomic components could be precisely classified with these techniques. Metabolomics technologies have been used previously to investigate plant drought responses and tolerance. For instance, Bowne et al. employed a targeted GC-MS approach to classifying compounds that vary in three bread wheat genotypes with varying levels of drought tolerance [26].

Even though extensive research has been done on the mulberry plant under drought stress using transcriptomics [9], little is known on the metabolomics approach in terms of revealing the mulberry’s metabolites under drought stress. Hence, this study aimed to investigate the metabolites profile of mulberry plant leaves under drought stress and control treatment using a non-targeted liquid chromatography-mass spectrometry (LC-MS) approach at a global level. In addition, this study will help to understand the drought-tolerant metabolites mechanisms involved in the mulberry variety Yu-711.

## 2. Results

### 2.1. Physiological Changes in Mulberry Yu-711 Response to Drought Stress and control Treatment

Physiological parameters, including relative water content (RWC) and leaf lengths of plants under drought stress and control, were examined before metabolites profiling to show how drought stress advances. The physiological evaluation reveals that mulberry leaf length remained shorter under the drought stress as the days for the drought period increased compared to the control treatment (Figure 1b). Under drought stress, leaf growth is mainly restricted due to stomata closure. On the other hand, the RWC of the drought-stress leaves decreased from 59.4% on the first day to 55% on the three days time point, and further down to 42% on the five days time point. Meanwhile, the control sample either increased or remained constant (Figure 1c).

### 2.2. Material Peak and Metabolite Statistics

A total of 20153 sample peaks and 8525 metabolites at both the ESI (−) and ESI (+) modes were detected from both the drought stress (EG) and control (CK) treatments. Out of the total peaks and metabolites, 8518 and 3165 were obtained from the negative ion mode. In the positive ion mode, 11635 peaks and 5360 metabolites were detected (Figure 2, Appendix A).

### 2.3. Metabonomic Changes in the Mulberry Leaves between EG_CK

The metabolites between the EG and CK were compared via LC-MS to identify the overall metabolomic changes in the mulberry leaves. Clean data were used for further analyses after the data had been filtered. An evaluation was made on the repeatability and stability of the metabolites using Principal component analysis (PCA), Partial least-squares-discriminant analysis (PLS-DA), and Orthogonal partial least-squares-discriminant analysis (OPLS-DA) score plots, which were used to analyze the similarities and differences between all samples associated with drought stress and controls (Figure 3). Using metabolite data from EG and CK, three main principal components (PCs) were created (Figure 3a), indicating that samples were classified into three groups on both (ESI) models, ESI (−) and ESI (+). Furthermore, samples from the same treatments were clustered together, indicating the quality of the data. Here, the first two PCs showed a total variation of 61.8%, with the principal component (PC1) explaining 48.8% of the total variation.

In contrast, 13% of the variation was explained by the second principal component (PC2) across the data set. This indicates that the changes in the metabolite profiles were caused by drought stress, and the differences in control samples were significant. The difference between drought stress and the control group (EG_CK) was detected and assessed using partial least square discriminant analysis (PLS-DA) in the ESI (+) or ESI (−) model, which yielded similar categorization results as in the PCA (Figure 3b). According to the treatment, samples were gathered in a certain area. The model’s prediction accuracy was validated using EG and CK data.

A total of 8523 metabolites were identified from 20153 material peaks (Appendix A). Amongst the metabolites are lipids and lipid-like molecules, phenylpropanoids and polyketides, organic acids and derivatives, organoheterocyclic compounds, nucleosides, nucleotides, and analogs. The lipid and lipid-like molecules, including prenol lipids, sphingolipids, glycerophospholipids, glycerolipids, fatty acyls and steroids, and steroid derivatives, were detected. In addition, phenylpropanoids and polyketides, including flavonoids, cinnamic acids, and derivatives, were also detected. Also, organoheterocyclic compounds, such as tetrapyrroles and derivatives, and pyridines and derivatives, were detected. Carbohydrates and carbohydrate conjugates in the class of the organooxygen compound and benzenoid compounds were expressed in response to the water deficit.

### 2.4. Analysis of the Main Differential Metabolites

The EG_CK analysis yielded a total of 945 metabolites via LC-MS that met the VIP criterion (VIP > 1) of the OPLS-DA model and had a significant *t*-test (*p* < 0.05). A total of 794 metabolites (84%) had no change. However, 100 (10.6%) and 51 (5.3%) metabolites were down-regulated and up-regulated, respectively (Figure 4a, Appendix A). Further analyses are revealed in Figure 4b, which indicates that drought stress or the control led to the differences in the metabolites. The heat map pattern of metabolites between samples and consistency among biological replicates is shown in Figure 4c. As a result, the 945 differentially expressed metabolites were classified based on their abundance (fold change) and weight (VIP) (Table 1, Appendix A).

The VIP values for metabolites categorized by superclass are shown in Figure 5a. The lipids and lipid-like compounds categories had the most weight once again. For example, the compound 2E,13Z-Octadecadienal of the lipids and lipid-like molecules recorded a VIP value of 19.331, even though the average mean of the exclusive VIP was 2.213. Figure 5b demonstrates that the most abundant differentially expressed metabolites, lipids, and lipid-like molecules, accounted for 47.7% of the total, while unclassified and organic oxygen compounds made up 19.2% and 10.2%, respectively.

Hierarchical cluster analysis (HCA) was performed to demonstrate the relative contents of the significant differentially abundant metabolites. Figure 6a shows the HCA heat map, whereas Figure 6b shows the histogram of the top 50 metabolites. Most of the altered metabolites were more abundant in EG. However, the majority had a low concentration in an abundance of the significantly expressed metabolites compared with those in the CK. This suggests that long drought stress led to more substantial fluctuations in the metabolites. Remarkably, Kaempferol 3-rhamnosyl-(1->6)-glucosyl-(1->6)-galactoside, epothilone A, thioridazine, 2-hydroxyhexadecanoic acid, tetramethylquercetin 3-rutinoside, D-maltose 2-hydroxyhexadecanoic acid, and neryl rhamnosyl-glucoside all increased significantly in EG, whiles 2E,13Z-octadecadienal, farnesyl acetone, and palmitic amide all decreased significantly in EG. The higher the VIP value, the greater the contribution. Figure 7 shows the 20 most differentially expressed metabolites with fold change values (Appendix A).

The relationships between EG and CK metabolites with significant differences were investigated using correlations analysis. Figure 8a demonstrates the Pearson correlation coefficients of the VIP values of the top 50 metabolites in EG and CK. The network interactions among metabolites are depicted in Figure 8b (Appendix A).

### 2.5. Metabolic Pathways Enrichment Analysis

Pathway enrichment analysis was performed using the KEGG (https://www.kegg.jp (accessed on 5 March 2021)) pathways database to investigate biochemical changes better. The differential metabolites were significantly enriched in metabolic pathways including arginine biosynthesis, galactose metabolism, alanine, aspartate, glutamate metabolism, ABC transporters, Linoleic acid metabolism, arachidonic acid metabolism, aminoacyl-tRNA biosynthesis, and glycerophospholipid metabolism. Figure 9a,b shows the top 20 metabolic pathways enriched by the differential expressed metabolites. Based on the enrichment analysis, 14 out of 64 pathways, including arginine biosynthesis, galactose metabolism, alanine, aspartate and glutamate metabolism, ABC transporters, linoleic acid metabolism were significantly enriched. In addition, those that were significantly enriched with *p*-values less than 0.05 are shown in Figure 9c,d (Table 2, Appendix A). Those involved in arginine biosynthesis, galactose metabolism, starch and sucrose metabolism, and arachidonic acid metabolism increased. In contrast, those involved in alanine, aspartate, and glutamate metabolism, linoleic acid metabolism, glycerolipid metabolism, carbon fixation in photosynthetic organisms, and arachidonic acid metabolism decreased, which aided in their function as energy storage.

## 3. Discussion

Drought is one of the most significant environmental factors limiting plant performance, development, and yield worldwide. The phenomenon causes many dramatic alterations in all plant organs on a morphological, physiological, and biochemical level, disrupting the sink and source plant organs [27]. Thus, a plant stress response is a complicated and dynamic process that aims to build new homeostasis in unfavorable growing conditions. More specifically, drought-responsive systems include hormone induction, kinase signaling, gene expression regulation, scavenging of reactive oxygen species, osmolyte production, cell structure modification, ion channel activation, carbohydrates and energy metabolism, nitrogen absorption, and amino acid metabolism, including fatty acid metabolism [28]. Thus, these processes involving genes, proteins, and small molecules (metabolites) all play a role in this dynamic process. However, metabolites are critical as they are directly involved in plant cell structure and metabolism, influencing the final phenotype [29]. Therefore, understanding the principles of stress adaption physiology and biochemistry, requiring a precise and simultaneous investigation of the metabolome in drought-tolerant and susceptible plant cultivars, is of great essence.

China is subjected to a variety of climatic circumstances, including low and high temperatures, as well as drought, all of which have an impact on agricultural productivity [6]. Mulberry cultivar Yu-711 exhibits distinct anatomical, morphological, and agronomic characteristics under natural conditions such as water stress. It is still unclear the metabolomic mechanism under which mulberry Yu-711 adapts to water stress. Metabolites regulate phosphorylation, acetylation, and peroxidation in various biological pathways [30]. Recent advances in metabolite identification and measurement have enhanced our understanding of this complicated process [30].

In this study, seedlings of mulberry variety Yu-711 were subject to two weeks of drought stress to investigate the metabolomics response to the water deficit and control using a non-targeted approach via LC-MS. The results show that lipids, organic acids, phenylpropanoid, organic oxygen compounds, organoheterocyclic, and other metabolites (Appendix A), and their pathway metabolism may play a role in drought stress response.

### 3.1. Lipids and Lipid-like Metabolites Change in Response to Drought Stress

Lipids are important energy storage molecules, as well as membrane components and signaling molecules [31]. Datasets from the LC-MS results in this study show that lipids and lipid-like metabolites were more prevalent after days of drought stress in mulberry Yu-711. Lipids constituted 451 of the total 945 differentially significant metabolites, accounting for 48% of the total differentially significant metabolites (Figure 5b). Only 65 lipids metabolites, accounting for 14.4%, were significantly changed in the drought stress leaves compared to the control. However, 386 lipids constituting 85.6% were unchanged in the drought stress leaves compared to the control (Appendix A). Water deficiency strongly altered lipids composition.

Major classes of lipids levels reveal a significant decrease under drought stress leaves compared to the control. This trend was prevalent for lipid classes such as fatty acyl, steroids and steroid derivatives, polyketides, sphingolipids, glycerolipids, glycerophospholipids, and sterol lipids (Appendix A). Interestingly, a significant increasing trend was observed for prenol lipids. Fatty acyl (FA) molecules and their derivatives were the most abundant lipids fractions produced in response to drought stress after the two-week drought period. Further investigation of the FA molecules revealed that 129 of the 192 metabolites were of low concentration, with lower VIP values in EG than CK (Appendix A), implying that the FA concentrations fell after drought stress treatment. Interestingly, 19 FA metabolites were significantly changed in the drought stress leaves compared to the control (Appendix A).

The significantly changed FA metabolites had a diverse trend, producing a 73.6% decrease. The most affected metabolites belong to the subclasses eicosanoids, linoleic acids, and fatty acids. The eicosanoids, including prostaglandin G2 ( PGG2), and 6-keto-PGF1alpha, decreased by 42.8%. In addition, linoleic acids (9(S)-HODE, 9(S)-HPODE, gamma-linolenic acid), as well as fatty acids ((4Z,7Z,10Z,13Z,16Z,19Z)-docosahexaenoate, 9,10,13-TriHOME, and traumatic acid), decreased by 21.4% each. On the other hand, a slight increase (23.4%) in FA content in the drought stress leaves compared to the control treatment was observed. However, eicosanoid was the most significantly affected, increasing by 40%. The substantial decrease in FA observed in mulberry could be due to membrane damage. Our results agree with the finding report from oat plant leaves [31]. Similar results in FA have been reported [32], which agrees with our results.

An increase in eicosanoid levels during drought stress in oat plants has been reported to help maintain membrane stability [33], which also agrees with our results. FA, including linolenic acid, is essential for membrane integrity and the functionality of important membrane proteins, such as those that make up the photosynthetic machinery; thus, a fall in its concentration significantly influences photosynthesis [31]. Furthermore, the degradation of linolenic acid and other polar lipids through hydrolytic processes could release free fatty acids (FFAs) and lipid hydroperoxides, which initiate the senescence process [34]. Prenol lipids (PR) content was the second most abundant after the fatty acyl after the drought stress period. Prenols are made from the five-carbon precursors isopentenyl diphosphate and dimethylallyl diphosphate, synthesized mainly by the mevalonic acid pathway [35,36].

Our results reveal that 97 of the 451 lipids metabolites (Appendix A), representing 21.5% (Appendix A), were PR. Only 17 prenol metabolites changed significantly. The significant change in PR metabolites exhibits a diverse trend with a 64.7% increase and a 35.3% decrease in the drought stress leaves compared to the control. Interestingly, the most affected metabolites are in the subclass of isoprenoids and terpenoids (Appendix A). Changes in PR levels have been reported in soybean [37] when subjected to drought treatment. The study demonstrated PR levels elevated during drought stress.

Similarly, our study also produced increased levels of PR in mulberry exposure to drought stress. Another study [38] indicated that the PR levels were increased due to the accumulation of ABA levels. ABA signaling plays a crucial role in plant physiology and helps plants respond to stressful environmental conditions [38]. In addition, PR plays a key role in the transport of oligosaccharides across a membrane [32]. Here, the accumulated level of the PR in the drought stress mulberry leaves could suggest a high level of ABA accumulation in the membrane during the drought period.

Glycerophospholipids (GPL) and glycerolipids (GL) are widespread and are important components of a cell’s lipid bilayer. These galactose lipids are a key component of biological membranes and provide intracellular and intercellular protein binding sites. Some GPL in eukaryotic cells are either precursors of membrane-derived second messengers or are second messengers themselves [32]. Water deficiency reveals a significant change in GPL and GL content in mulberry leaves. In this study, 47 of the 451 were in the GPL class representing 10.4% of the total lipids, whiles 14 of the total lipids, representing 3.1%, were GL (Appendix A). Even though most of the GPL and GL content decreased in the drought stress leaves, they were significantly not changed from the control.

Interestingly, eight and one metabolite(s) changed significantly in the GPL and GL, respectively, under drought stress (Appendix A). The GPL decreased by 75% and a slight increase of 25%. However, the most affected metabolites were glycerophosphocholines (PC) and glycerophosphates. On the other hand, the glycosylglycerols of GL decreased by 100%. According to [37], GPL metabolite 1-(sn-Glycero-3-phospho)-1D-myoinositol was at a lower concentration when soybean was exposed to drought. In this study, the 1-(sn-glycero-3-phospho)-1D-myoinositol level decreased under drought stress, which is consistent with those obtained in soybean leaves [37].

The decrease in galactose lipids and phospholipids content in thyme plant drought stress [32] is consistent with our results. Even though some of the GPL increased, they were not significant from the control treatment, which agrees with the report [30,39]. The increased PC and glycerophosphoethanolamines (PE) in the current study agree with those obtained previously [40]. PC and PE are part of major cell membrane lipids, which act as an osmoprotective compound, indicating that these metabolites act as a protective mechanism against osmotic stress in the mulberry plant [41].

Exposure of the mulberry plant to drought stress reveals other lipids classes such as steroid and sterol and their derivates, polyketides (PKs), sphingolipids, and saccharolipids (Appendix A). In this study, 35 PK metabolites consisting of a subclass of flavonoid, aromatic polyketides, macrolides, and lactone polyketides were identified (Appendix A). Interestingly, only 4 PKs metabolites changed significantly under water deficit. However, the PKs level increased by 75%. The most affected compounds are macrolides, lactone polyketides, and aromatic polyketides (Appendix A). PKs are typically responsible for the biosynthesis of essential secondary metabolites involved in plant defense and signal transduction [42].

Sphingolipids form major structural components of the plasma membrane and other endomembrane systems. These lipids are indispensable cellular constituents that aid in signal transduction under drought conditions [43]. Plants can maintain a stringent equilibrium between nonphosphorylated and phosphorylated sphingolipids to govern the cell’s destiny under stress [43]. Sphingolipids (mainly free LCBs) exist in cells at very low concentrations [44]. Our result shows that eight metabolites of the sphingolipids class of lipids were detected with a decrease in content (Appendix A); however, only one sphingolipid metabolite changed significantly under water deficit (Appendix A). The amounts of the various sphingolipid classes in plants differ by species and tissue [45]. In this study, the level of GlcCer was the highest sphingolipids, forming 37%, however, there was no significant change under the drought and control treatment. An increasing level of sphingolipids has been reported in plants under water deficit; their results gave evidence about the involvement of sphingolipids in drought-related signaling [46]. Here, our result supports those obtained in the thyme plant [32].

Our results indicate that 53 of the 451 lipids metabolites were steroids and sterols. However, only 15 of them had a significant change. The steroid and sterols content decreased significantly by 86.6%, and a slight increase of 13.3% under the drought stress leaves compared to the control was observed (Appendix A). Phytosterols are essential components of a plant’s membrane lipid bilayer as they protect plants against stresses and control the membrane’s fluidity, thus affecting its properties, functions, and structure [47]. Some studies have reported increased phytosterol during drought stress [32,48]. However, our study reveals a decreased level of phytosterol. As an adaptive response to environmental conditions, high terpene synthase (TPS) activity levels increase sterol content [49]. The terpene level in this study was higher, and TPS might have enhanced the sterol content. The reduced phytosterol level in this study indicates that the mulberry under stress conditions activates a distinct pathway as a defense to the stress treatments and which have led to forming other derivatives with a more substantial water holding capacity. The results here agree with the study reported by [49].

### 3.2. Organic Oxygen Metabolites Change Response to Drought Stress

Drought stress strongly affected the level of organic oxygen when the mulberry plant was exposed to drought stress, accounting for 10.2% (Figure 5b) of the total differential metabolites. The metabolites were mainly carbohydrates and their conjugates, and an exciting trend of total carbohydrate metabolites was observed. Generally, higher (40) and lower (41) carbohydrate metabolites concentrations were detected under the drought leaves compared to the control (Appendix A). However, further analysis shows that most of the sugars were not significantly changed. Remarkably, only 21 organic oxygen metabolites changed significantly (Appendix A). The most affected compounds are carbohydrates, alcohols, and polyols, decreasing at 65%. Among these compounds are beta-cortol, chlorogenic acid, fructose 6-phosphate, fucose 1-phosphate, glucose 6-phosphate, N-Acetyl-D-glucosamine, N-Acetyl-D-glucosamine 6-Phosphate, and quinic acid.

On the other hand, an increased level of organic oxygen, mainly carbohydrates, recorded a 35% increase under the drought stress leaves. This trend was evident in metabolites such as 3-methoxy-4-hydroxyphenylglycol glucuronide, alpha-Santalyl acetate, D-(+)-raffinose, D-maltose, levan, maltotriose, and octanoylglucuronide. Intercellular CO_2_ (Ci) decreases due to stomatal closure during moderate water deficit while photosynthetic capacity is maintained. Reversible inhibition of some enzymes, such as sucrose-phosphate synthase (SPS), may occur due to the decrease in Ci. Simultaneously, starch content falls, whereas reducing sugars remain constant or even rise [50]. This change in the carbohydrate status can alter cellular processes such as gene expression [50]. Sugar deprivation causes significant physiological and biochemical changes in order to maintain respiration and other metabolic functions [51]. However, sugar concentration and source-sink partitioning are not affected according to distinctive patterns in different organs and under different stresses [51,52,53].

Generally, under drought and other stresses such as salinity, soluble sugar concentrations increase. In contrast, sugar concentration is reduced under high light irradiance (PAR, UVBR), heavy metals, nutrient shortage, and ozone [51]. Nonetheless, sugar changes vary with genotype and stress factors and do not follow a static model [54]. Our result is similar to those reported on the diverse trend of sugar changes under water deficit [55]. For example, maltose, an essential sugar that helps plant growth and development, was reported to increase after a water deficit [56]. Also, some reports have revealed that the accumulation of sugar metabolites such as glucose, fructose, and raffinose levels elevated and were the earliest metabolites accumulated after withholding water to induce water deficits [57,58]. Remarkably, not all soluble sugars play the same function in events involving metabolites of stressed plants. Sucrose and glucose, for example, are either cellular respiration substrates or osmolytes that maintain a cell’s homeostasis [59]. However, fructose is unrelated to osmoprotectant and synthesizes secondary metabolites [59].

According to [60], fructose may be linked to the synthesis of erythrose-4-P, which serves as a substrate to produce lignin and phenolic compounds. All this indicates that soluble sugar metabolism is a dynamic process involving degrading and synthesizing reactions under stressful conditions [60]. The carbohydrate status of the leaf, which is affected by water deficit in quantity and quality, could function as a metabolic signal in the stress response [61], even though the signaling role of sugars is still not fully understood. Plant sugar signaling is part of a complex network that also includes a plant-specific hormone signaling pathway. Glucose-6-phosphate (G-6-P) is reported to be involved in repression signals. However, a decrease in intracellular concentration after treatment with glucose reveals that G-6-P is involved in the direct signal [62]. Thus, the low concentration level of G-6-P in the results may suggest the crucial signal pathway role of G-6-P in the mulberry under water deficit.

### 3.3. Phenylpropanoid and Polyketide Metabolites Change Response to Drought Stress

The phenylpropanoid and polyketides metabolites content was significantly altered, accounting for 8.1% (Figure 5b) of the total differential metabolites when mulberry seedlings were exposed to drought stress. However, the metabolites were mainly involved in the class of flavonoids (38.9%), and cinnamic acid derivatives (23%) (Appendix A). Generally, there was a dynamic trend in the polyphenols expressed on the mulberry plant under the water deficit. Interestingly, 40 of the metabolites were of low concentration, while 38 of them had an increased concentration. However, most of them did not change significantly. Polyphenols such as Quercetin 3-O-(6″-malonyl-glucoside) 7-O-glucoside, apigenin 4′-[feruloyl-(->2)-glucuronyl-(1->2)-glucuronide] 7-glucuronide, 2′-hydroxygenistein 7-(6″-malonylglucoside), 2′,7-Dihydroxy-4’-methoxy-8-prenylflavan 2′,7-diglucoside, and Kaempferol 3-sophorotrioside were strongly altered under water deficit; however, they were significantly not changed. Interestingly, only 14 of the polyphenols changed significantly. These metabolites mainly involve flavonoids, cinnamic acids, and their derivatives subclass (Appendix A). In the present study, the polyphenols content increased at 57.2% against a 42.8% decrease under the drought stress. Flavonoids are antioxidant phenolic compounds, an important class of secondary metabolites found in plants that protect them against various stressors. Secondary metabolites are synthesized from phenylalanine through a conserved pathway in plants [57]. There have been several reports on phenolic compound alteration in plants under drought stress. For example, an increase in the flavonoid compound in Arabidopsis under drought stress has been reported [55]. Our result is similar to those obtained by these authors. Flavonoid responses in times of drought stress may serve as reactive oxygen species (ROS) scavengers in the vacuole.

Cinnamic compound, another important phenylpropanoid compound, was significantly altered. Thus, drought stress has a reducing effect on cinnamic content. According to [49], a reduced cinnamic acid was recorded when strawberry plants were exposed to drought stress. Similarly, our results produced a decrease in cinnamic content. For instance, 2-hydroxycinnamic acid and m-Coumaric acid were mostly affected with lower concentrations. Cinnamic acid may play a role by regulating the flux into the phenylpropanoid pathway [49]. Sun and colleagues exposed cucumber to water deficit and found that the level of cinnamic acid was reduced [63].

Further, downstream from cinnamic acid has been found in mulberry fruits, described as a potential antioxidant compound [64]. Plants under stress have been observed to accumulate phenylpropanoids. Their role in combatting this stress may be linked to scavenging free radicals in their function as a stress response marker.

### 3.4. Organic Acids Metabolic Changes to Drought Stress

The exposure of mulberry plants to drought stress altered the organic acid metabolites contents, constituting 5.4% of the total differential metabolites. The metabolites involved were mainly in the class of carboxylic acid and their derivatives (90%), comprising a subclass of amino acids, peptides, and their analogs (Appendix A). The accumulation of amino acids under drought stress has been described in many plant species [55]. In this study, 51 organic acid metabolites were expressed under drought stress. Further analysis reveals that 39 metabolites of the organic acids were altered, but they were not changed significantly under the drought stress compared to the control. Among them were, 13E-tetranor-16-carboxy-LTE4, arginyl-phenylalanine, ceanothine D, 2-(3-carboxy-3(trimethylammonio)propyl)-L-histidine, and kinetensin 1-3. Conversely, only 12 metabolites of organic acid changed significantly (Appendix A).

Many plant species have been reported to accumulate amino acids in response to drought stress [55]. The amino acid altered decreased by 81% and the most affected included branch chain amino acids (BCAA) and the aspartate family. Amino acids play a crucial role in drought stress by acting as an osmolyte, regulating ion transport, modulating stomatal opening, enzyme synthesis, and affecting gene expression and redox-homeostasis [26,58].

Branch chain amino acids (BCAA) such as valine, leucine, isoleucine, and other related amino acids are reported to generally increase in response to drought stress [55,58]. An increased isoleucine level has been reported in rice under drought stress [65]. Conversely, the isoleucine content in the present study decreases significantly under the drought stress compared to the control. BCAAs are known to accumulate in relatively large concentrations under stress; therefore, it is uncertain if they could act as suitable solutes [66]. However, under water-deficit stress, alternate respiration pathways have been demonstrated to boost the utilization of BCAAs. Importantly, in Arabidopsis, the alternate pathway of respiration and enhanced BCAA catabolism, but not their accumulation, contributed to drought tolerance [58]. The significantly decreased level of isoleucine could suggest the involvement of the compound in the scavenging of ROS.

Free amino acid, such as arginine, has been shown to provide osmotic adjustment in plants under drought stress by inhibiting stomatal opening [67]. An increased level of arginine has been reported in rice leaves after drought stress [68]. Also, [69] reported high arginine content in peanuts after drought stress. Conversely, in our results, arginine content significantly decreased in drought leaves compared to the control. Nevertheless, drought stress does not always cause an increase in amino acid content. Moral and colleagues reported decreased arginine content when the wheat plant was grown under drought stress [70]. They argue that the level of amino acid accumulation could depend on genotype and duration of drought stress. Our result agrees with their findings as almost all the significantly changed amino acids decreased.

Histidine is an essential amino acid required by the plant for growth and development. It also acts as a metal-binding ligand and is a vital component of the metal hyperaccumulator molecule, mitigating heavy metal stress [71]. Its role in drought stress has been implicated, including aiding in the stomatal opening [67,71]. Under drought stress, histidine concentration is said to increase [71]. However, our result shows that histidine decreased under water stress leaves compared to the control. Water deficit increases the generation of reactive oxygen species (ROS). Maintaining or increasing the activity of enzymes involved in eliminating harmful ROS to minimize cellular damage is thought to be a key element in dehydration tolerance [71]. Thus, the low content of histidine may suggest the compound’s active role in scavenging the ROS to maintain cellular integrity.

The aspartate family, comprising aspartic acid and asparagine, plays a crucial role in plants against drought stress. Accumulation of aspartate family, notably asparagine in plants during water deficit, has been reported [72]. The authors reported that the concentration of asparagine increased throughout the drought period from 4 to 10 days in bread wheat. Interestingly, our result revealed that all the aspartate families decreased in concentration, which agrees with the observation made in other plant species [27].

### 3.5. Other Metabolic Changes to Drought Stress

Other superclasses of metabolites, including organoheterocyclic compounds, benzenoids, nucleosides, nucleotides, analogs, lignans, neolignans, and related compounds, were notably detected. Remarkably, organoheterocyclic compounds and benzenoids were among the most differentially regulated compounds during drought stress. From the present study, metabolites in the superclass of organoheterocyclic compounds only formed 4.6% (Figure 5b) of the total differentially detected metabolites. However, this compound class was one of the most significantly changed metabolites in the drought leaves compared to the control (Appendix A).

Further analysis reveals that 18 metabolites of organoheterocyclic compounds significantly changed. Remarkably, adenine, aminophylline, guanine, milrinone, 1-Deoxynojirimycin, and fagomine are among those that decrease significantly (Appendix A). In addition, metabolites such as 2-Methyl-1-hydroxypropyl-ThPP, limonene-1,2-epoxide, riboflavin, mefloquine, and topotecan increased significantly. Major purine bases, such as adenine and guanine, are essential compounds in activating tolerance mechanisms in protecting nucleic acids [27]. These compounds provide the ultimate energy source for synthesizing carbohydrates, lipids, peptides, and secondary metabolites [73]. An exposure of spring wheat to drought stress increased adenine and guanine content [27]. On the contrary, we observed a decrease in content which clearly shows that the purine and metabolites were significantly altered when the mulberry plant was exposed to drought stress. Interestingly, a down-regulated purine level has been reported in soybeans after exposure to drought and heat stress [73], which agrees with our results.

The piperidines class of compounds, including fagomine and 1-Deoxynojirimycin (1-DNJ), are essential organoheterocyclic compounds. These iminosugars have been isolated from the mulberry plant, which plays a vital role in Chinese traditional medicine [74,75]. Several reports on iminosugars have been on their biological function in treating various human diseases. In the present study, mulberry exposure to drought stress revealed a significant decrease in the content of 1-DNJ and fagomine, with 1-DNJ having a fold change of −1.689 and a VIP value of 14.682, and fagomine having −1.909 and a VIP of 4.430. The content of 1-DNJ varies based on some factors such as environment, location, and mulberry variety.

High 1-DNJ and fagomine content have been determined in the mulberry plant [76]. According to [74], 1-DNJ content in the mulberry (Yu-711) plant was lowest when different mulberry varieties from different locations and weather conditions were studied, which agrees with the current result. During drought stress, the sugar level is altered due to the limitation on CO_2_ assimilation. Thus, the low level of iminosugars may suggest their active role in scavenging ROS during the drought period and may act as an osmoprotectant. The results show that drought stress may have a negative effect on iminosugar content in the mulberry plant.

Riboflavin (vitamin B_2_) is an essential compound for plant growth and development. Riboflavin belongs to the pteridines class, and its role in drought stress has been reported [77]. Our result revealed that the content of riboflavin was up-regulated in the mulberry plant after drought stress. The fold change in the compound was 0.660, with a VIP value of 1.651. Thus, riboflavin at small and moderate quantities has been inferred to enhance plant drought tolerance, while very high content impaired drought tolerance in plants [77]. According to Deng and colleagues, high levels of riboflavin in the plant cause an accumulation of ROS. Interestingly, in our results, the content of riboflavin increases in the drought leaves compared to the control, and the magnitude was less. This may suggest that the lower extent of the compound in the drought leaves might have enhanced the plant tolerance to the drought period by scavenging the ROS.

Metabolites in the benzenoid superclass were significantly altered in the drought leaves compared to the control (Appendix A). From the results, 12 compounds in the class belonging to naphthalenes, phenols, and benzene substitutes and derivatives under the benzenoid superclass were significantly altered in the drought leaves compared to the control (Appendix A). Remarkably, compounds such as hippuric acid, p-salicylic acid, quercetin 3-(6″-malonyl-glucoside), and isoproterenol are amongst the compounds significantly decreased (down-regulated) in the drought leaves; however, 2-hydroxy-2-[4-hydroxy-3-(3-methylbut-2-en-1-yl)phenyl] acetic acid, metoprolol, and 6-paradol significantly increased (up-regulated) under the drought leaves compared to the control.

Salicylic acid (SA), a plant hormone, is a promising compound crucial in plants’ sensitivity to environmental stresses through regulating the antioxidant defense system, transpiration rates, stomatal movement, and photosynthetic rate [78]. Furthermore, exogenous SA application has been reported to enhance plant tolerance to drought stress by reducing ROS activity, thereby improving membrane stability [79]. Interestingly, SA content in our results decreased in the drought stress leaves compared to the control. This may suggest that during the drought stress, the biosynthesis of SA was used to scavenge ROS activity to enable membrane stability for the plant to endure the drought conditions.

On the other hand, quercetin 3-(6″-malonyl-glucoside) content was far lower in concentration from the drought leaves than the control with a fold change (log2FC) of −34.294 and a VIP of 5.056. Quercetin is a crucial polyphenol compound that acts as a plant antioxidant during abiotic stress. Mulberry plants have been reported to contain many polyphenols(flavonoids) [80]. The low level of quercetin 3-(6″-malonyl-glucoside) suggests its activeness against ROS, thus maintaining membrane integrity.

### 3.6. Metabolic Pathway Analysis

The differential metabolites pathway map indicates that some essential metabolites were significantly involved in the drought stress response. For example, in the arginine biosynthesis pathway, metabolites such as L-aspartic acid (C00049), L-arginine (C00062), argininosuccinic acid (C03406), and oxoglutaric acid (C00026) were significantly involved in the pathway (Appendix A). These metabolites (carboxylic acids and derivatives class) significantly decreased except oxoglutaric acid (Keto acids and derivatives), which increases in response to the drought stress. The decrease in amino acid and an increased oxoglutaric acid content involved in drought stress response has been reported [81], which agrees with our results suggesting the role of organic acid in drought stress management. Metabolites involving lipid and carbohydrates were significantly altered (Appendix A). Drought stress has a significant influence on lipid and sugar content in plat response to such stress. The lipid and sugar metabolites involved in the pathway include galactosylglycerol, alpha-Santalyl acetate, D-(+)-Raffinose, and beta-cortol.

Alanine, aspartate, and glutamate metabolism pathways reveal significant changes in key metabolites involved in the pathway during drought stress. These metabolites comprise oxoglutaric acid, L-aspartic acid, L-asparagine, and argininosuccinic acid (Appendix A). In the linoleic acid metabolism pathway, 9(S)-HODE, gamma-linolenic acid, 9(S)-HPODE, and 9,10,13-TriHOME changed significantly in response to the water stress (Appendix A). Similar results on lipids changes [32] in plants response to drought stress effect have been reported. In the ABC transporters, key metabolites such as L-aspartic acid, L-arginine, D-glycerol 1-phosphate, L-histidine, N-acetyl-D-glucosamine (GlcNAc), D-maltose, and riboflavin were altered significantly in response to drought stress (Appendix A). The increased riboflavin content in this work in managing ROS agreed with a report on other plant species when exogenous riboflavin was studied under drought stress [77].

Some metabolites were enriched in multiple transporters. For instance, L-arginine (C00062) is involved in various transporters such as phosphate and amino acid transports. A significant change was observed in metabolites affecting aminoacyl-tRNA biosynthesis during the drought stress in the plant response to the water deficit. These metabolites are L-aspartic acid, L-arginine, L-histidine, L-isoleucine, and L-asparagine (Appendix A). The changes in these amino acids may suggest their enhancement role in the plant cell during drought stress [67]. They play various functions such as ion transport across the membrane, stomatal inhibition, and opening during drought stress.

The arachidonic acid metabolism pathway comprising 20-hydroxy-LTE4, 20-hydroxy-leukotriene B4, 6-keto-PGF1alpha, prostaglandin G2 (PGG2), traumatic acid, and 9S,11R,15S-trihydroxy-2,3-dinor-13E-prostaenoic acid-cyclo[8S,12R] metabolites changed significantly during the drought stress and were enriched in this pathway (Appendix A). In addition, key metabolites such as D-glycerol 1-phosphate, PA(0:0/18:2(9Z,12Z)), phosphocholine, glycerylphosphorylethanolamine(PA), and lysoPC(17:0) were significantly enriched in the glycerophospholipid metabolism pathway during the drought stress (Appendix A). According to a study, stress-induced alterations in lipid class profiles might trigger membrane lipid remodeling and activate plant defense responses to biotic and abiotic stresses such as drought [82]. The significant changes in the membrane lipids in the present study agree with the research reported in other plant species [32]. Also, in the carbon fixation in photosynthetic organisms pathway, L-aspartic acid, fructose 6-phosphate, and D-sedoheptulose 7-phosphate were altered significantly in response to the water deficit (Appendix A). The decreased level of these compounds in the pathway suggests their role in scavenging ROS to protect the plant from membrane damage.

In the cyanoamino acid metabolism pathway, L-asparagine, L-aspartic acid, L-isoleucine, and dhurrin were significantly involved during the water deficit. Interestingly, all the metabolites involved in the pathway significantly reduced in concentration (Appendix A). A decrease in the amino acid content in this pathway may suggest that these compounds protected the plant membrane from excess water loss by causing stomatal inhibition during the water stress [67]. On the other hand, the content of dhurrin, and cyanogenic glucoside, which serves as a defensive compound in sorghum [83], was significantly altered in response to the water deficit, which may suggest its membrane defensive role in the mulberry plant. The histidine metabolism pathway was significantly enriched with oxoglutaric acid, L-aspartic acid, L-histidine, and 4-imidazolone-5-propionic acid (Appendix A).

In addition, the beta-alanine metabolism pathway comprises L-aspartic acid, L-histidine, and pantothenic acid, which were significantly enriched in the pathway during the drought stress (Appendix A). The starch and sucrose metabolism route significantly altered fructose 6-phosphate, glucose 6-phosphate, and D-maltose (Appendix A). Finally, through mulberry exposure to water deficit, the glycerolipid metabolism pathway also showed significant changes in D-glycerol 1-phosphate, PA(0:0/18:2(9Z,12Z)), and galactosylglycerol (Appendix A).

## 4. Materials and Methods

### 4.1. Source of Mulberry Plants and Growth Conditions

The mulberry species (*Morus alba*) Yu-711 was obtained from the National Mulberry Gene Bank in Jiangsu University of Science and Technology, Zhenjiang, Jiangsu, China. Plant growth was as per [6] and is described below. Mulberry plants were grown in a greenhouse with a 14 h light/10 h dark photoperiod, at 25 °C Day/20 °C night temperature, and relative humidity of 70–80%. The cuttings were grafted to the rootstocks. The grafted nurseries, reaching the three-leaf stage, were planted in pots of 35 cm diameter containing loam soil with one seedling per pot. A total of 18 pots were grouped into two (drought and control), with each group containing nine pots for a total of three replicates with three pots each. The control and drought groups were watered daily until when new shoots had grown to 20 cm.

### 4.2. Drought Stress Treatment

Upon developing the seedlings in the pots when the fresh leaves have emerged, the drought stress experiment commenced. Water supply was withdrawn for 14 days in the experimental group to induce natural drought stress. However, the control group was constantly supplied with water daily. Once the experimental seedlings reached the wilting point (symptoms apparent), leaf sampling was done. Drought-related characteristics such as leaf relative water content (RWC) and leaf length were measured prior to sampling. The first three-time point for sampling after 14 days of drought stress was the first day (1 day), the third day (3 days), and the fifth day (5 days). To account for diurnal variations in metabolite levels, control and drought-treated plants were sampled simultaneously (midday). The primary leaf tissue samples were harvested and immediately frozen in liquid nitrogen and then stored at −80 °C. Leaves from experimental and control groups from the 5 days time point (n = 16) were collected and pooled for metabolite extraction and analysis.

### 4.3. Leaf Length Measurement

Determination of leaf measurement was as described by [84]. Leaf length of all the seedling samples was measured after 1 days, 3 days and 5 days time points. This was achieved by using a ruler to measure the length of the leaf, starting from the tip to the sheath on about 18 plants at each time point. The data obtained from the leaf measurement were subjected to a two-tailed Student *t*-test using GraphPad Prism 9 to determine the significant variation (*p* < 0.05) between the control and drought-treated group. The data are presented on a figure as means ± SD of three biological replicates (three plants per replicate).

### 4.4. Relative Water Content Evaluation (RWC)

The seedling RWC was evaluated to assess the impact of drought treatments. We evaluated the RWC as described by [84]. In summary, paper bags were baked at 65 °C for approximately days when a constant weight was attained. The wait of fresh leaves was determined (WF) and soaked in distilled water for 24 h. The leaves were weighed again to determine their saturated weight (WFT) and then fixed for 30 minutes at 105 °C. After that, the leaves were placed in the dried paper bags and incubated for three days at 80 °C. The constant dry weight (WD) was determined using three independent samples. The RWC was determined from the following formula: RWC = (WF − WD)/(WFT − WD) × 100%. The significant difference between the control and the drought group was determined by a two-tailed Student *t*-test (*p* < 0.05) using GraphPad Prism 9. The data from three biological replicates is represented on a figure as means ± SD.

### 4.5. Metabolite Extraction

All chemicals and solvents used were analytical or HPLC grade. Water, methanol, acetonitrile, and formic acid were purchased from CNW Technologies GmbH (Düssel-dorf, Germany).

Düsseldorf, Germany. However, L -2-chlorophenylalanine was obtained from Hengchuang Biotechnology Co. Ltd Shanghai, China. The metabolites were extracted as described [85,86]. Eight replicates (n = 16) were analyzed in each group of mulberry (YU-711). In brief, samples were ground at 60 Hz for 2 min and ultrasonicated at ambient temperature (25–28 °C) for 10 min. After centrifuging at 10,000× *g* at 4 °C for 15 min, supernatants were dried using a freeze-concentration centrifugal dryer and then resuspended in methanol and water (1:4, v:v), vortexed for 30 s, incubated at 4 °C for 2 min, and centrifuged at 10,000× *g* at 4 °C for 5 min. Finally, the supernatants were filtered through 0.22 µm microfilters, transferred to LC vials, and stored at t 80 °C for LC-MS analysis.

### 4.6. Mass Spectrum Data Calling

An Acquity UHPLC (Waters Corporation, Milford, MA, USA) was coupled with an AB SCIEX 5600 TripleTOF System (AB SCIEX, Framingham, MA, USA) (Q Exactive Orbitrap, Thermo Fisher Technologies, Waltham, MA, USA). The conditions for the LC-MS were followed as described previously [85].

### 4.7. Data Filtering and Analysis

The acquired LC-MS raw data were analyzed by the progenesis QI v2.3 software (Nonlinear Dynamics, Newcastle, UK) using the following parameters: precursor tolerance was set at 5 ppm, fragment tolerance was 10 ppm, and retention time (RT) tolerance was set at 0.02 min. Internal standard detection parameters were deselected for peak RT alignment, isotopic peaks were excluded for analysis, and noise elimination level was set at 10.00. The minimum intensity was set to 15% of base peak intensity. Any peaks with missing values (ion intensity = 0) in more than 50% of samples were removed, further reducing the resulting matrix. An internal standard was used for data quality control (QC). The QC samples were prepared by mixing aliquots of all samples into a pooled sample, injected at regular intervals (every 10 samples) throughout an analytical run to provide a set of data in which repeatability could be assessed. Progenesis QI v2.3 software (Nonlinear Dynamics, Newcastle, UK) for data processing was used to identify the metabolites using public databases from http://www.hmdb.ca/; http://www.lipidmaps.org/ (accessed on 5 March 2021).

Principal component analysis (PCA) and (orthogonal) partial least squares discriminant analysis (O)PLS-DA analysis [87] were carried out on the combined positive and negative ion mode data to visualize the metabolic alterations among experimental groups to identify the differentially expressed metabolites. In addition, the variable importance in the projection (VIP), which is a weighted sum of PLS loading that is widely used to find the most important characters in metabonomic data [88], was employed to rank the overall contribution of each variable to the OPLS-DA model and those variables with VIP >1 and *p* < 0.05 (two-tailed Student *t*-test) were considered as differentially expressed metabolites. Also, the 7-fold cross-validation and response permutation test (n = 200) was adopted in validating the model (Figure 3d). The R^2^ and Q^2^ validation plot (Figure 3d) indicates that the model was reliable without overfitting.

The correlations between the VIP levels of metabolites in EG and CK were investigated using Pearson’s correlation coefficients. In addition, metabolic pathway enrichment using metabolites mapped to the Kyoto Encyclopedia of Genes and Genomes (KEGG) database (http://www.kegg.jp/ (accessed on 5 March 2021)). was used further to investigate the relative abundance of differentially expressed metabolites. A hypergeometric test was used to determine the *p*-value of each metabolic pathway derived to find significantly enriched pathways [30] and false discovery rate (FDR) correction, and the metabolic pathways with *p* < 0.05 were retained. Furthermore, compound names were uploaded to pathway analysis through MetaboAnalys [89]. Finally, the precise pathway analysis algorithms of the hypergeometric test were used in mapping to the pathway library of the *Arabidopsis thaliana* KEGG database (in real-time).

## 5. Conclusions

Investigations on the effects of various abiotic stresses, such as drought, on relevant food and medicinal plants, are important for economic and educational purposes, as well as decision making. This will undoubtedly assist plant growers and breeders in expanding their knowledge of these plants’ stress tolerance mechanisms. As a result, they will be able to increase their yield under stressful situations. In this regard, we investigated metabolomics changes at the global level caused by drought stress in the mulberry variety Yu-711 plant for the first time. According to our findings, the leaves of the mulberry variety Yu-711 exposed to water deficiency displayed considerable alterations in numerous metabolite classes. Total lipids, galactolipids (MGDGs and DGDGs), and phospholipids (PG, PE, PA, and PS) decrease generally in stressed plants. Fatty acyl lipids were the most abundant lipids produced. Prenol lipids, on the other hand, increased significantly in the drought-stressed plant compared to the control. Organic oxygen, precisely carbohydrates, had dynamic changes with decreased and increased patterns under the water deficit. In addition, polyphenols (mainly antioxidant secondary metabolites including flavonoids and cinnamic acids) also produced a dynamic change leading to both increasing and decreasing patterns under the drought-stressed plant.

A significant change in organic acids, particularly amino acids, was a significant decrease in content following the drought stress. In addition, other classes of metabolites, including benzenoid and organoheterocyclic, decreased significantly under water deficit.

Metabolic pathway analysis also reveals that some essential metabolites were significantly involved in the drought stress response. For example, these metabolites were significantly involved in the pathways including arginine biosynthesis, galactose metabolism, alanine, aspartate, glutamate metabolism, ABC transporters, Linoleic acid metabolism, arachidonic acid metabolism, aminoacyl-tRNA biosynthesis, and glycerophospholipid metabolism during the drought stress period. As a result, we can speculate that droughts drive the biosynthesis of structural membrane lipids and other metabolite types in mulberry Yu-711 plants to protect the cell and chloroplast membranes from severe drought damage and preserve their structure and function. Furthermore, increasing quantities of antioxidants and flavonoids in these plants’ levels would also help scavenge ROS generation under droughts.

Finally, other molecular approaches, such as integrated transcriptomics and metabolomics, will undoubtedly help us to understand the genes behind the phenomena observed, providing new insight into adaptation mechanisms for mulberry plant response to drought stress conditions in the current global climate change situation.

## Figures and Tables

**Figure 1 plants-10-01636-f001:**
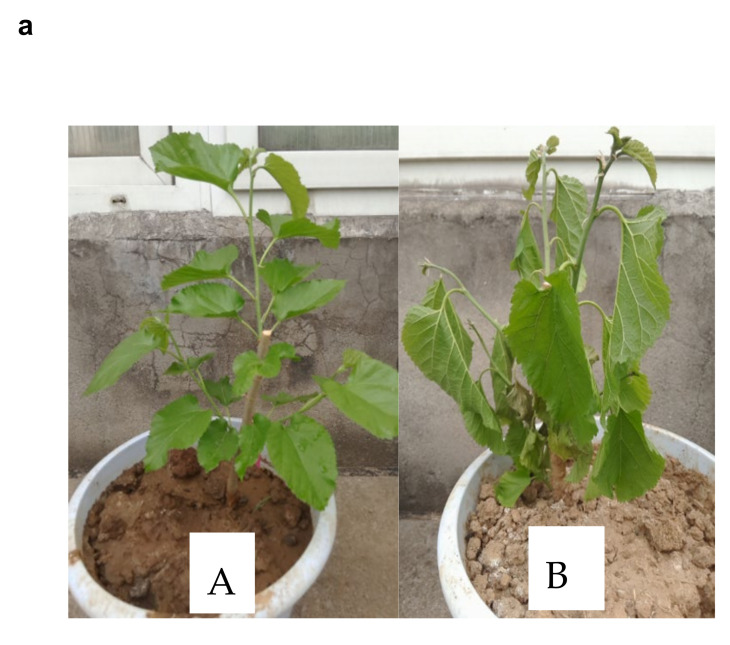
Physiological responses of seedling leaves affected by drought and the control treatment. (**a**) Mulberry seedlings under drought and control experimental setup. (Panel A); seedlings under control treatment. (Panel B); seedlings under drought stress treatment at the five days time point. (**b**) Leaf length measured at one days, three days, and five days time points. (**c**) The relative water content at one days, three days, and five days time points. The values in a and b are represented as mean ± SD of three replicates with three plants per replicate. The asterisks (***) and (****) denote a significant and highly significant difference, respectively (*p* < 0.05) according to a two-tailed Student *t*-test using GraphPad Prism 9. CK; Control treatment group. EG; Drought stress treatment group.

**Figure 2 plants-10-01636-f002:**
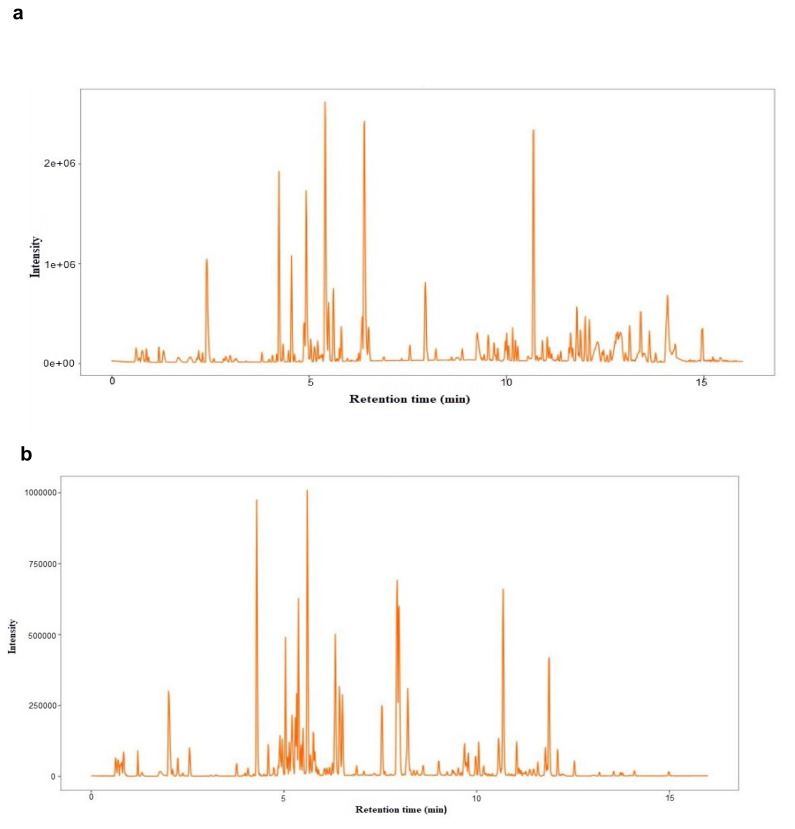
Sample peaks and metabolites. (**a**) Peaks at the positive ion mode. (**b**) Peaks at the negative ion mode. (**c**) Bar chart of all sample peaks and the corresponding metabolites in both the negative and positive mode. (**d**) Metabolite intensity distribution between EG, CK, and QC. QC; quality control.

**Figure 3 plants-10-01636-f003:**
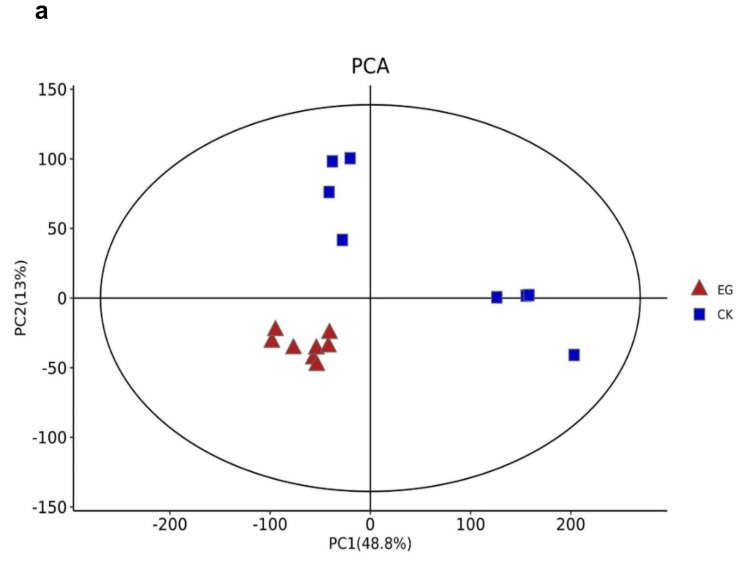
The difference in metabolites between EG and CK in mulberry (Yu-711) leaves based on multivariate statistical analysis. (**a**) Principal component analysis (PCA). (**b**) Partial least-squares-discriminant analysis (PLS-DA). (**c**) Orthogonal partial least-squares-discriminant analysis (OPLS-DA). (**d**) A 200 times permutation test of OPLS-DA mode. R^2^ = (0.0, 0.766); Q^2^ = (0.0, -0.598).

**Figure 4 plants-10-01636-f004:**
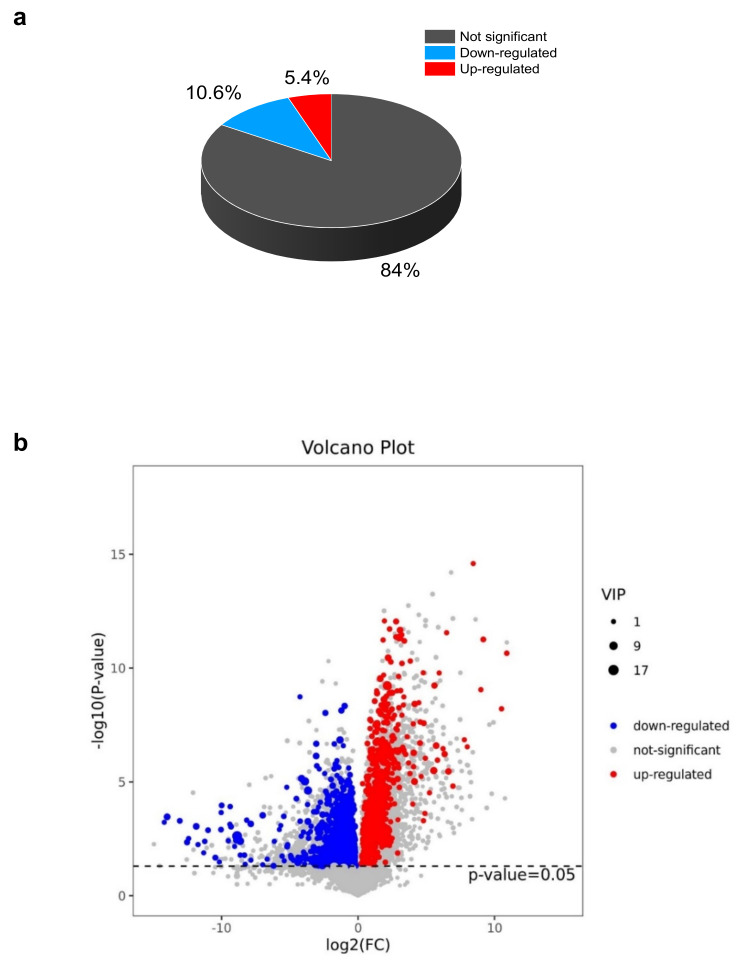
The fold change (FC) arrangement and pattern of the main differential metabolites in EG_CK in Yu-711 leaves. (**a**) The proportion of differential metabolites in EG_CK. (**b**) Volcano plot on metabolites that were significantly different in EG_CK. The red color indicates metabolites with high concentrations. The blue color is metabolites with low concentrations. The grey color indicates no change in different metabolites. (**c**) Pattern heat map between samples. The color legend, which ranges from blue to red, reflects the abundance of metabolites, ranging from low to a high concentration.

**Figure 5 plants-10-01636-f005:**
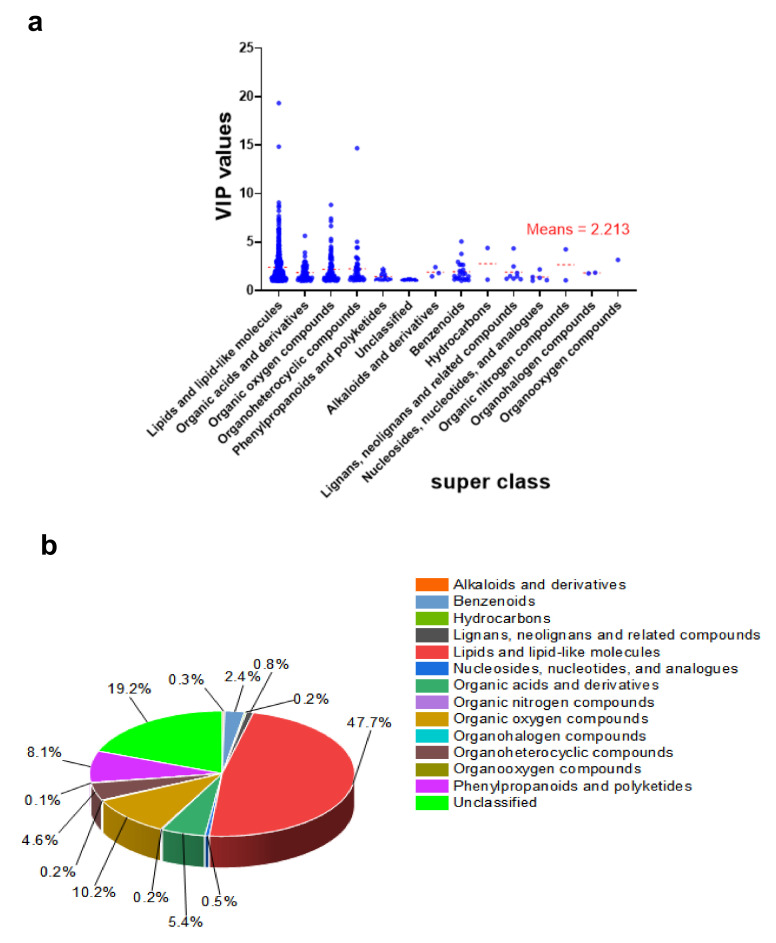
Classification of the differentially expressed metabolites and their variable importance in the projection (VIP) distribution in EG_CK in mulberry Yu-711 leaves. (**a**) VIP distribution in each metabolite superclass as a scatter plot. The average mean of the differentially expressed metabolites is 2.213, and the red dashed line is the individual means. (**b**) A pie chart depicting the proportion of each metabolite in the superclass. Lipids and lipid-like molecules represent 47.7%, followed by unclassified at 19.2%, phenylpropanoid and polyketides at 8.1%, and organic oxygen compounds at 10.2%.

**Figure 6 plants-10-01636-f006:**
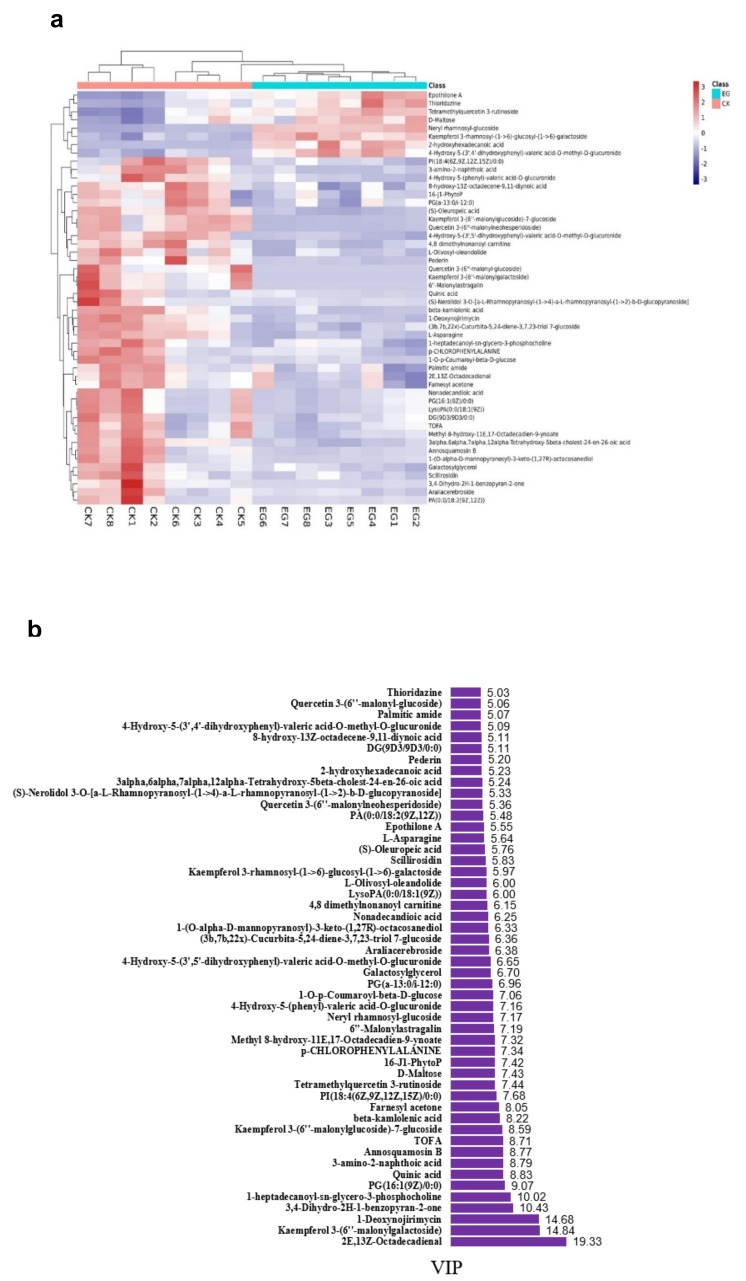
Heat map of the top 50 differentially expressed metabolites in EG CK in mulberry Yu-711 leaves based on hierarchical clustering analysis. (**a**) Differentially expressed metabolites separated by hierarchical clustering. The x-axis depicts (1–8) biological replicates of each type of treatment sample, and the y-axis represents the differentially expressed metabolites separated by hierarchical clustering. From blue to red color indicates an increase in metabolites abundance from low to high concentration; (**b**) VIP values of each differentially expressed metabolite.

**Figure 7 plants-10-01636-f007:**
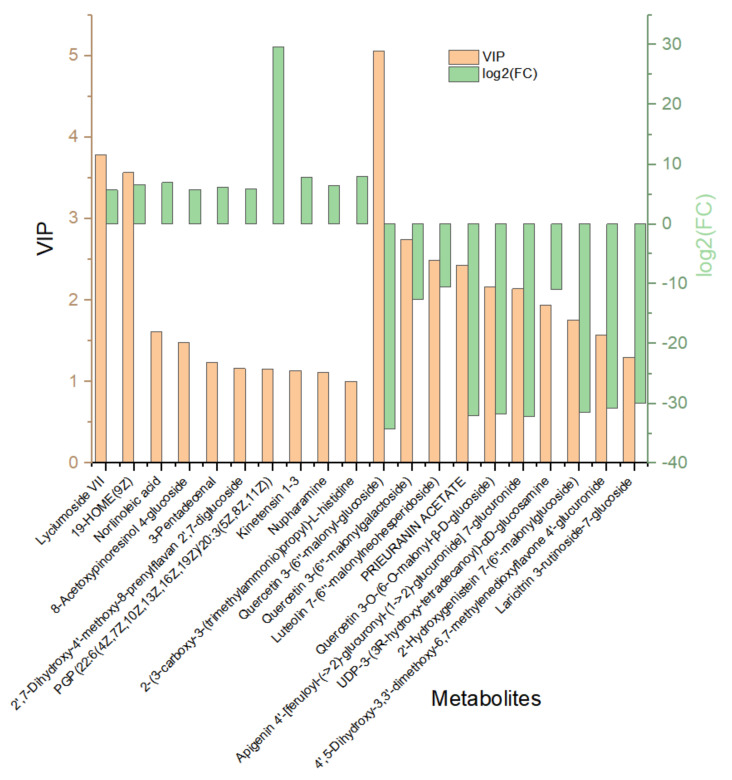
Bar graph of 20 differentially expressed metabolites in EG_CK in mulberry Yu-711 leaves. VIP values in a brown column and the green columns represent log2 (fold change, FC) values. The left half represents high concentrated metabolites, while those in the right half are low in concentration.

**Figure 8 plants-10-01636-f008:**
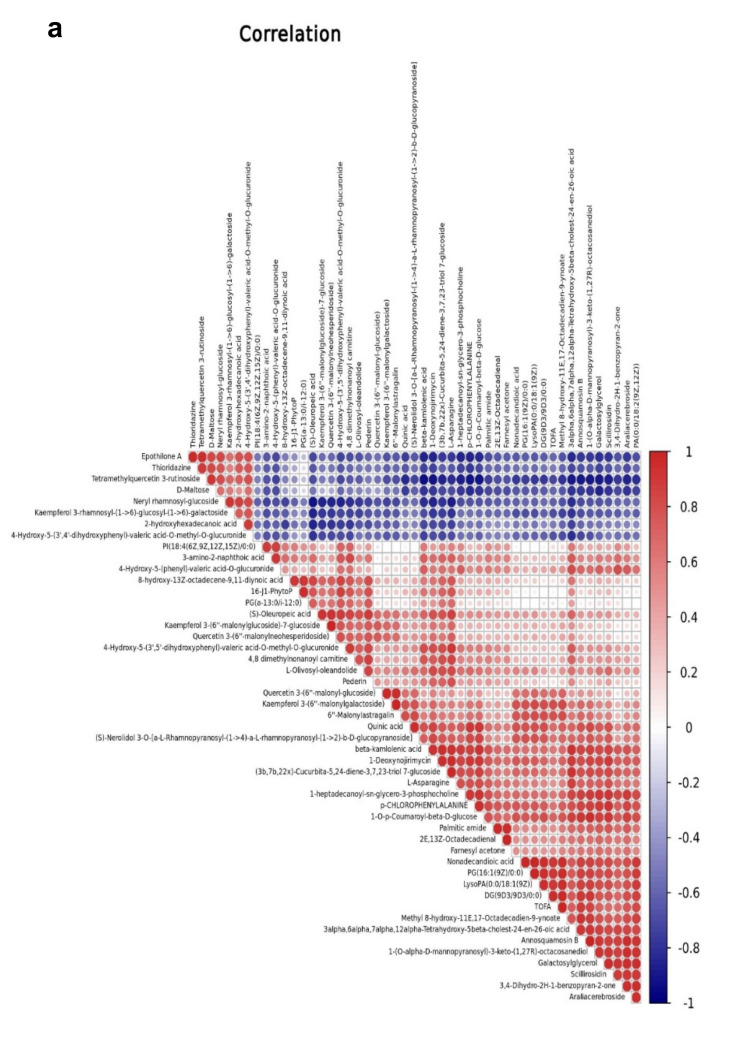
Correlation analysis and network of differentially expressed metabolites in EG_CK in the mulberry Yu-711 leaves. (**a**) Pearson’s coefficients of correlation of VIP values depicting the relation among metabolites between EG and CK. Positive correlation in red and blue is the negative correlation. Different sizes of circles indicated the correlation of Pearson’s coefficients. (**b**) Interactions between classes of the top 50 differentially expressed metabolites in a network. The threshold for Pearson’s correlation coefficient was set at 0.9. Positive and negative correlations between compounds are represented by red and blue lines, respectively.

**Figure 9 plants-10-01636-f009:**
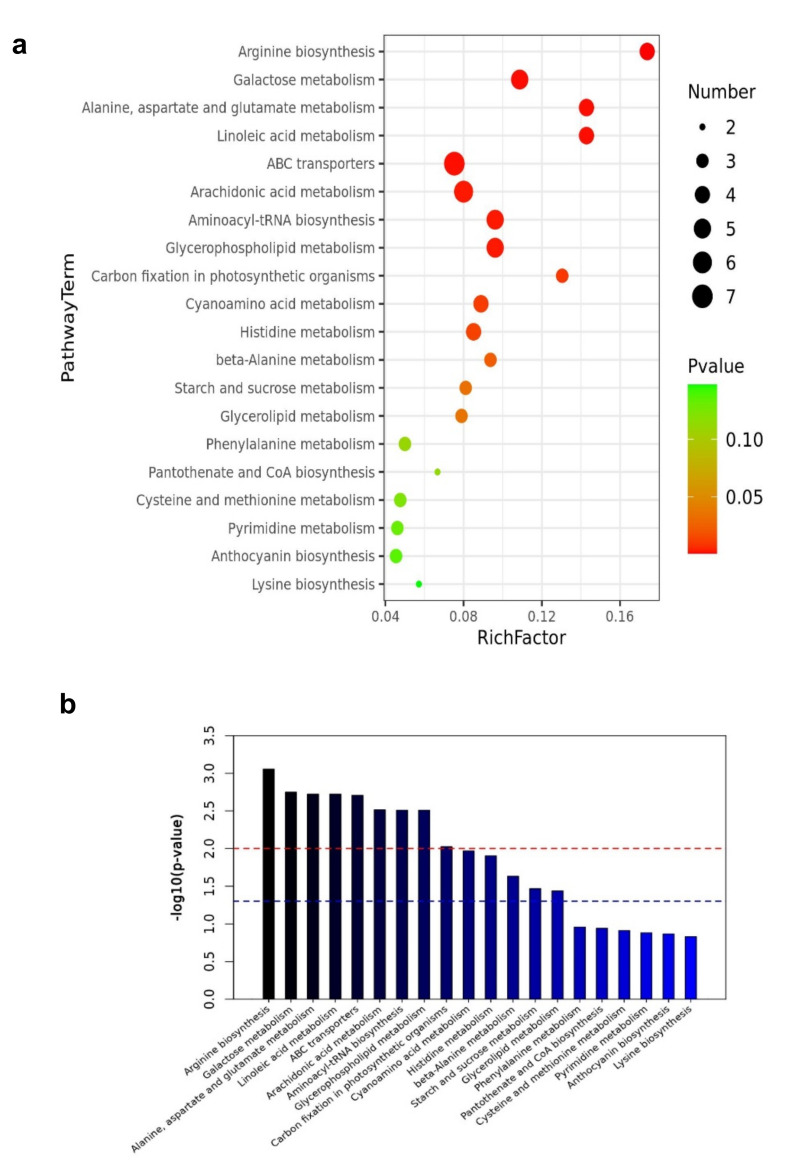
Pathway enrichment analysis of the differentially expressed metabolites of EG_CK in mulberry Yu-711. (**a**) Analysis of the top 20 metabolic pathways enriched by differential expressed metabolites using a heat map. The analysis focused on metabolic pathways visualization analysis obtained from KEGG (ttp://www.kegg.jp/ (accessed on 5 March 2021)). The color from green to red denotes that the *p*-value decreases sequentially. The size of the bubble represents the number of metabolites enriched in each pathway. (**b**) Bar graph showing that the *p*-value of the top 20 metabolites involved in the metabolic pathway is significant. The red dash line indicates the *p*-value = 0.01, and the blue dash line indicates the *p*-value = 0.05. The top of the bar above the blue line means the signal pathway represented by it is significant. (**c**) Heat map analysis of the top 20 metabolic pathways with a *p*-value less than 0.05 enriched by differentially expressed metabolites. From green to red indicates *p*-value decreases sequentially; the point size indicates the number of metabolites enriched in each pathway. (**d**) A bar graph showing the *p*-value of not more than 0.05 of significant metabolites involved in the pathway. The red and blue dash line means *p*-value = 0.01, and 0.05, respectively.

**Table 1 plants-10-01636-t001:** Top 50 different metabolites with VIP values of all significantly different metabolites in EG_CK of Yu-711 leaves.

RT (Min)	Ion Mode	Metabolites	VIP	*p*-Value	Adj. *p*-Value(FDR)	log2(FC)
13.030	pos	2E,13Z-Octadecadienal	19.33	0.02	0.05	−0.44
5.538	pos	Kaempferol 3-(6″-malonylgalactoside)	14.84	0.00	0.01	−8.86
0.712	pos	1-Deoxynojirimycin	14.68	0.00	0.00	−1.69
2.717	pos	3,4-Dihydro-2H-1-benzopyran-2-one	10.43	0.03	0.07	−3.00
11.472	pos	1-heptadecanoyl-sn-glycero-3-phosphocholine	10.02	0.01	0.04	−0.40
13.987	pos	PG(16:1(9Z)/0:0)	9.07	0.02	0.05	−1.45
0.824	neg	Quinic acid	8.83	0.04	0.08	−1.67
4.052	pos	3-amino-2-naphthoic acid	8.79	0.00	0.01	−2.91
12.609	pos	Annosquamosin B	8.77	0.01	0.02	−2.12
11.065	pos	TOFA	8.71	0.01	0.04	−1.18
4.704	pos	Kaempferol 3-(6″-malonylglucoside)-7-glucoside	8.59	0.00	0.00	−3.89
11.267	pos	beta-kamlolenic acid	8.22	0.00	0.01	−1.13
12.304	pos	Farnesyl acetone	8.05	0.03	0.06	−0.51
13.527	pos	PI(18:4(6Z,9Z,12Z,15Z)/0:0)	7.68	0.01	0.04	−1.12
5.595	pos	Tetramethylquercetin 3-rutinoside	7.44	0.00	0.01	1.19
0.841	neg	D-Maltose	7.43	0.02	0.05	0.37
8.883	pos	16-J1-PhytoP	7.42	0.01	0.04	−0.34
4.145	pos	p-CHLOROPHENYLALANINE	7.34	0.02	0.05	−0.30
12.115	pos	Methyl 8-hydroxy-11E,17-Octadecadien-9-ynoate	7.32	0.01	0.02	−1.45
5.528	neg	6″-Malonylastragalin	7.19	0.00	0.02	−8.64
5.772	pos	Neryl rhamnosyl-glucoside	7.17	0.00	0.00	3.04
5.557	pos	4-Hydroxy-5-(phenyl)-valeric acid-O-glucuronide	7.16	0.00	0.02	−1.23
4.546	pos	1-O-p-Coumaroyl-beta-D-glucose	7.06	0.02	0.04	−1.28
12.490	pos	PG(a-13:0/i-12:0)	6.96	0.03	0.06	−0.40
0.831	pos	Galactosylglycerol	6.70	0.01	0.04	−0.43
3.092	pos	4-Hydroxy-5-(3’,5’-dihydroxyphenyl)-valeric acid-O-methyl-O-glucuronide	6.65	0.00	0.00	−1.78
15.079	pos	Araliacerebroside	6.38	0.04	0.07	−0.66
11.513	pos	(3b,7b,22x)-Cucurbita-5,24-diene-3,7,23-triol 7-glucoside	6.36	0.00	0.00	−0.94
14.449	pos	1-(O-alpha-D-mannopyranosyl)-3-keto-(1,27R)-octacosanediol	6.33	0.01	0.04	−1.90
14.007	pos	Nonadecandioic acid	6.25	0.02	0.05	−1.23
5.480	pos	4,8 dimethylnonanoyl carnitine	6.15	0.00	0.00	−1.29
14.002	neg	LysoPA(0:0/18:1(9Z))	6.00	0.02	0.04	−1.61
5.195	pos	L-Olivosyl-oleandolide	6.00	0.00	0.01	−1.00
4.585	pos	Kaempferol 3-rhamnosyl-(1->6)-glucosyl-(1->6)-galactoside	5.97	0.00	0.00	1.33
10.195	pos	Scillirosidin	5.83	0.02	0.05	−0.41
2.403	pos	(S)-Oleuropeic acid	5.76	0.00	0.00	−1.32
0.712	pos	L-Asparagine	5.64	0.00	0.00	−2.19
5.933	pos	Epothilone A	5.55	0.00	0.00	1.47
12.629	pos	PA(0:0/18:2(9Z,12Z))	5.48	0.04	0.08	−2.88
4.706	neg	Quercetin 3-(6″-malonylneohesperidoside)	5.36	0.00	0.00	−4.14
10.157	pos	(S)-Nerolidol 3-O-[a-L-Rhamnopyranosyl-(1->4)-a-L-rhamnopyranosyl-(1->2)-b-D-glucopyranoside]	5.33	0.05	0.09	−2.89
11.886	pos	3alpha,6alpha,7alpha,12alpha-Tetrahydroxy-5beta-cholest-24-en-26-oic acid	5.24	0.00	0.00	−0.99
12.493	neg	2-hydroxyhexadecanoic acid	5.23	0.00	0.00	5.55
5.195	pos	Pederin	5.20	0.00	0.01	−2.44
14.071	pos	DG(9D3/9D3/0:0)	5.11	0.03	0.07	−0.99
10.724	pos	8-hydroxy-13Z-octadecene-9,11-diynoic acid	5.11	0.00	0.01	−0.26
2.738	pos	4-Hydroxy-5-(3’,4’-dihydroxyphenyl)-valeric acid-O-methyl-O-glucuronide	5.09	0.00	0.00	1.66
12.778	pos	Palmitic amide	5.07	0.02	0.05	−0.46
5.270	pos	Quercetin 3-(6″-malonyl-glucoside)	5.06	0.00	0.02	−34.29
8.364	neg	Thioridazine	5.03	0.00	0.01	1.81

RT (min) represents retention time. Pos/neg means the metabolite profiling obtained via ESI positive or ESI negative ion modes. VIP is the variable important in projection. FDR: Force detection rate represented by adj. *p*-value.

**Table 2 plants-10-01636-t002:** Analysis of metabolic pathway enrichment based on a *p*-value of less than 0.05.

ID Annotation	Annotation	Match Status	RichFactor	*p*-Value	−lg (*p*-Value)	FDR Correction
ath00220	Arginine biosynthesis	4/23	0.173913043	0.00088	3.055582081	0.024688537
ath00052	Galactose metabolism	5/46	0.108695652	0.001781	2.749220415	0.024688537
ath00250	Alanine, aspartate, and glutamate metabolism	4/28	0.142857143	0.00189	2.723430403	0.024688537
ath00591	Linoleic acid metabolism	4/28	0.142857143	0.00189	2.723430403	0.024688537
ath02010	ABC transporters	7/93	0.075268817	0.001956	2.708576464	0.024688537
ath00590	Arachidonic acid metabolism	6/75	0.08	0.003057	2.514704648	0.024688537
ath00970	Aminoacyl-tRNA biosynthesis	5/52	0.096153846	0.003086	2.510594625	0.024688537
ath00564	Glycerophospholipid metabolism	5/52	0.096153846	0.003086	2.510594625	0.024688537
ath00710	Carbon fixation in photosynthetic organisms	3/23	0.130434783	0.009444	2.024862056	0.067154545
ath00460	Cyanoamino acid metabolism	4/45	0.088888889	0.010732	1.969316695	0.068685218
ath00340	Histidine metabolism	4/47	0.085106383	0.012479	1.90380436	0.072607742
ath00410	beta-Alanine metabolism	3/32	0.09375	0.023355	1.631620415	0.124559918
ath00500	Starch and sucrose metabolism	3/37	0.081081081	0.034158	1.466512782	0.167226148
ath00561	Glycerolipid metabolism	3/38	0.078947368	0.036581	1.436747753	0.167226148

Match Status = Hit/Total. The total is the total number of compounds in the pathway; the hit is the matched number. The *p*-value is originally calculated from the enrichment analysis; the −lg (*p*-value) is the *p*-value adjusted by logarithm operation based on 10; the FDR is the *p*-value adjusted using False Discovery Rate results from the pathway analysis.

## Data Availability

The original data sets described in the study are included in the article/Appendix A. Further inquiries can be addressed to the corresponding author.

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
