# Peer review of "Metabolomics Response to Drought Stress in Morus alba L. Variety Yu-711"

_plants, 2021, doi:10.3390/plants10081636_

Round 1

Reviewer 1 Report

The purpose of this work sound interesting and aimed to address important questions about the metabolic adaptation of an economically important crop to drought stress, an environmental factor that often negatively impacts on the final yield.  Overall, the quality of the analysis seems relatively OK, however, there are no data on the plants used for the study. So, all the results are speculative since the authors do not show to us if the plants effectively perceived a drought stress. Beside this, there are since modifications to apply to the figures.

Major points:

Upon reading the manuscript, the results section directly falls into the description of the metabolic analysis and avoid to explain the experimental setup and how the plants were obtained (add a figure). This point is crucial because the authors have to show some results to prove that plants effectively received a drought stress while the control did not… For example, a comparison of the fresh and dry weight and the water content is clearly mandatory. In addition, the authors have to add some measurements of water potential, since only this variable is valuable to determine and follow the application of a drought stress. For instance, the authors used the visual “wilting point “, a parameter which is highly doubtful and prone to errors/misinterpretations. Without these data, the present manuscript is purely speculative!

The analysis of the figure 1 clearly shows some big variations inside the group “CK” that are mainly explained by the Principal Component 2 of PCA, PLS-DA and OPLS-DA. However, the metabolites associated with this PC2 were all considered together for further analysis. This clearly introduce a strong bias, that should be corrected.

L19: The “numerous differentially-accumulated metabolic elements” are not as a function of time, since the authors only showed the results for the sampling point “5 days”…This is clearly a FALSE statement to boost the manuscript…Please remove this.

The resolution of all figures is clearly below 300 dpi and must be improved, given the number of information presented in each chart/heatmap/PCA.

The figure 4 should be enlarged to help the reader to read the legend.

Reviewer 2 Report

This is a fine descriptive manuscript. However, there are several points the authors should address:

  • It looks loke as if in the PCA analysis (also visible of course in the heat map) that the treated groups behaves more uniformly as the control group (two set of 4 samples of the control group cluster together). One would expect an outcome the other way round. Following the same line: the authors harvested samples at 1 d, 3 d, and 5 days after stressing the plants. Why did they analyze only the 3 d samples? Analyzing all of them would not have been significantly more work. Also the analysis of the samples immediately after the stress period (0 d without recovering) would have been interesting.
  • I am missing any information about FDR. Obviously in DDA acquisition based global metabolite profiling some identified metabolites are wrong (or isomers). Some assessment should be made or at least a comment about that problem made in the text. There is no public information available about the “self-made” database, posing a problem for somebody who would like to verify the results.
